# Urea Detection in Phosphate Buffer and Artificial Urine: A Simplified Kinetic Model of a pH-Sensitive EISCAP Urea Biosensor

**DOI:** 10.3390/s25216596

**Published:** 2025-10-26

**Authors:** Karen Simonyan, Astghik Tsokolakyan, Vahe Buniatyan, Artem Badasyan, Mkrtich Yeranosyan

**Affiliations:** 1Innovation Center for Nanoscience and Technologies, A.B. Nalbandyan Institute of Chemical Physics NAS RA, P. Sevak 5/2, Yerevan 0014, Armenia; karensim248@gmail.com (K.S.); astsokola@gmail.com (A.T.); mkrtich.yeranosyan@gmail.com (M.Y.); 2Materials Research Laboratory, University of Nova Gorica, Vipavska 13, 5000 Nova Gorica, Slovenia; 3Institute of Physics, Yerevan State University, A. Manoogian 1, Yerevan 0025, Armenia

**Keywords:** kinetic model, data fitting, steady-state approximation, urea biosensor, urease, artificial urine

## Abstract

A simplified kinetic model for the quantitative analysis of a potentiometric, pH-based urea biosensor is presented. The device was an electrolyte–insulator–semiconductor capacitor (EISCAP) with a pH-sensitive Ta_2_O_5_ gate functionalized by a polyallylamine hydrochloride (PAH)/urease bilayer. Within the steady-state approximation, the kinetic equations yielded an implicit algebraic relation linking the bulk urea concentration to the local pH at the sensor surface. Numerical solution of this equation, combined with a fitting routine, provides the apparent Michaelis–Menten constant ($K_{M}$) and the normalized maximum reaction rate (${\bar{k}}_{V}$). Validation against the literature data confirmed the reliability of the approach. Experimental results were then analyzed in both phosphate buffer (PBS) and artificial urine (AU), covering urea concentrations of 0.1–50 mM. The fitted parameters showed comparable $K_{M}$ values of 10.9 mM (PBS) and 32.4 mM (AU), but strongly different ${\bar{k}}_{V}$ values: $2.2\times 10^{-4}$ (PBS) versus $8.6\times 10^{-7}$ (AU). The three-order reduction in AU was attributed to the inhibitory effects inherent to complex biological fluids. These findings highlight the importance of the model-based quantitative analysis of EISCAP biosensors, enabling the accurate characterization of immobilized enzyme layers and guiding optimization for applications in realistic sample matrices.

## 1. Introduction

Urea is a key biomarker of renal and hepatic function, making its accurate detection in biological fluids essential for disease diagnosis and patient monitoring [1]. Non-invasive, point-of-care biosensors have gained significant attention for this purpose, offering rapid and user-friendly analysis compared with traditional blood tests. In particular, potentiometric enzyme biosensors provide high sensitivity and specificity with real-time results [2], and they can operate in easily obtainable sample matrices like urine or saliva [3]. Most urea biosensors exploit the enzyme urease to catalyze urea hydrolysis, detecting the resulting species—ammonium (NH_4_^+^), carbon dioxide (CO_2_) bicarbonate, or proton (pH) changes—that correlate with the initial urea concentration [4]. These devices encompass a variety of transducer types including potentiometric electrodes and field-effect transistor-based sensors, optical biosensors (e.g., colorimetric assays that measure urease-generated NH_4_^+^ via the classical Berthelot reaction) [5], and others. Indeed, urease-based biosensors have been developed using virtually every major detection principle: in addition to the pH-sensitive potentiometric and field-effect devices already noted, researchers have reported conductometric sensors (monitoring the increase in solution conductivity as urease produces ionic species) and amperometric biosensors (which couple the urease reaction to redox processes for current measurement) [3]. Each approach offers distinct advantages and limitations; for example, conductometric designs are simple but can be affected by background ionic strength, optical methods (such as Nessler’s or Berthelot’s reagent-based assays) achieve high accuracy but require multiple reagents, and amperometric setups often need added mediators or applied potentials since urea itself is not electroactive. Comparative studies indicate that electrochemical techniques—particularly potentiometric urease electrodes—tend to provide the best overall performance in terms of rapid response, stability, and ease of use. Meanwhile, the *gold-standard* for urea quantification in clinical laboratories remains an enzymatic colorimetric assay (the blood urea nitrogen test) that uses urease to generate ammonia, followed by a secondary reaction (e. g. with Berthelot’s or diacetylmonoxime reagent) to produce a measurable color change. Although highly accurate, these conventional methods demand dedicated reagents, trained personnel, and benchtop instrumentation, and typically require invasive blood sampling—factors that make them time-consuming and impractical for frequent or point-of-care use [3]. By directly converting the biochemical reaction into an electrical signal without additional reagents, enzyme-based biosensors enable faster, on-site measurements. They are robust, portable, and amenable to miniaturization—attributes that are particularly advantageous in resource-limited settings where conventional lab tests may be impractical [6]. This combination of sensitivity and practicality underpins the growing interest in urea biosensors for decentralized diagnostics.

Among the various urea biosensing approaches, field-effect devices have seen notable advancements. In particular, the electrolyte–insulator–semiconductor capacitive (EISCAP) sensor has emerged as a powerful platform [7,8,9,10,11,12,13,14,15]. The pH-sensitive EISCAP used in this work consists of an Al/p–Si/SiO_2_/Ta_2_O_5_ structure functionalized with a urease-rich layer (polyallylamine hydrochloride (PAH) and urease), which transduces enzymatic pH shifts at the gate interface into measurable capacitance or voltage changes [7,8]. EISCAP sensors are popular due to their simple design and straightforward, cost-effective fabrication (no photolithography or complex encapsulation required) [9,10]. Tantalum pentoxide (Ta_2_O_5_), used as the gate insulator, offers excellent chemical stability and near-Nernstian pH sensitivity (~59 mV per pH) [11], enhancing the transduction of urease-catalyzed pH changes. As a result, the EISCAP platform achieves high sensitivity and a wide dynamic range for analytes like urea [12]. Notably, Welden et al. employed EISCAP sensors for the measurement of acetoin in wine and beer media, demonstrating the versatility of this platform biochemical sensing application [13]. Compared with many existing urea sensors and benchtop assays, the reported EISCAP technology provides a unique combination of specificity (through the bio-selective urease layer), sensitivity (through the highly pH-responsive Ta_2_O_5_ transducer), and operational convenience. In summary, it enables reagent-free and real-time urea detection with excellent stability [13,14,15,16], addressing several limitations of conventional methods.

An important step toward real-world application is to validate the biosensor in complex sample matrices. Here, we assessed the sensor’s performance not only in standard phosphate buffer (PBS), but also in artificial urine (AU), a synthetic urine analog. AU has a controlled composition (containing urea, creatinine, uric acid, etc.) and a stable pH, making it a valuable tool for the calibration and validation of urea sensors while avoiding the variability of real human samples [17,18]. This testing allows us to examine potential interference by other substances present in a urine-like environment. In practical scenarios, various compounds in biological fluids can affect the local pH or enzyme activity, which may in turn influence the sensor’s response. By using AU, we can identify and quantify such matrix effects under controlled conditions. Our approach is broadly applicable: with appropriate calibration, the pH-sensitive EISCAP sensor could be deployed for urea monitoring in other liquid-phase systems beyond PBS and AU including actual clinical samples (e.g., human urine or saliva) [3], and potentially even in environmental or industrial water samples where urea analysis is relevant.

The growing demand for reliable and sensitive urea detection has also driven extensive efforts to develop mathematical models that describe the complex physicochemical processes within biosensors. For potentiometric pH-based biosensors modified with immobilized enzymes, the sensor’s response arises from an intricate interplay of biochemical and electrochemical reactions. This response is influenced by numerous factors including the enzyme immobilization method and layer geometry [5,19], the catalytic efficiency and enzyme–substrate interaction mechanisms [10,11], the buffer composition, pH, stirring rate [20,21], and ambient temperature as well as the concentration gradients between the sensing surface and the bulk solution [12]. Experimental approaches alone are often insufficient to isolate and quantify the individual effects of these parameters. Therefore, a deeper understanding of biosensor behavior requires integrating experiments with theoretical models. Such models can estimate key characteristics including the actual reaction rate (v), the Michaelis–Menten constant (K_M_), the maximum reaction rate (V_max_), and the rate constants for substrate, product, and buffer transport.

Mathematical modeling offers insights into experimentally inaccessible aspects of enzyme-catalyzed reactions such as the nature of transition states, intermediate species, and rate-limiting steps. It also helps identify catalytically significant interactions and other factors that affect biosensor performance. By estimating apparent kinetic and transport parameters like K_M_ and V_max_, modeling can elucidate their dependence on sensor conditions (e.g., pH, buffer strength, or temperature). Thus, modeling is an indispensable tool for predicting biosensor response under varying conditions and for enabling the rational design and optimization of sensor configurations.

The aim of this study was to quantitatively analyze the kinetic and transport parameters of an enzyme-modified Al/p–Si/SiO_2_/Ta_2_O_5_ EISCAP urea biosensor (featuring a PAH/urease bilayer) in both PBS and AU across a range of urea concentrations. We employed a simplified kinetic model developed previously [22,23,24,25,26,27,28] to derive an explicit steady-state relationship between the bulk urea concentration and the local pH at the sensor surface in each medium, under controlled reaction conditions. The predicted pH-versus-[urea] curves were then fit to our experimental data [22] (see Appendix A), yielding the apparent Michaelis–Menten constant and maximum reaction rate in PBS vs. AU. By integrating modeling with experimentation, this approach not only provides accurate concentration readouts, but also reveals how the enzyme kinetics are altered in complex fluids, highlighting the advantages of our EISCAP biosensor for quantitative urea analysis in realistic conditions.

## 2. Materials and Methods

### 2.1. Diffusion-Kinetic Model Theory

Traditionally, models describing enzymatic electrodes have focused on diffusion-dominated processes [29]. These models often rely on simplifying assumptions, such as treating enzymatic kinetics as first- or zero-order, in order to make the resulting equations solvable. However, the inherently nonlinear nature of the Michaelis–Menten equation leads to parabolic differential equations that are analytically intractable in general cases. Solutions are typically expressed in terms of complex integrals or power series expansions [30,31], and in some instances, extrapolation methods are employed instead [32].

Diffusion models are particularly suited for thick enzyme layers with well-defined geometries, where mass transport becomes the limiting factor. These models usually neglect the influence of solution stirring, although in practice, stirring significantly affects the sensor responses. To reconcile this, some approaches assume that diffusion coefficients within the enzyme layer vary with stirring rate [30,31], an assumption shared by other studies on electrodes with liquid-state membranes [33].

A more advanced strategy introduces the concept of a hypothetical interfacial zone between the enzyme layer and bulk solution, accounting for concentration gradients across this boundary [34]. While more comprehensive, this model requires specialized numerical techniques due to its mathematical complexity.

Most of the above studies address potentiometric sensors that are sensitive to enzymatic reaction products. When these diffusion-based approaches are adapted for pH-based sensors, the equations become more intricate. They involve a combination of diffusion equations with those governing protonation equilibria [35,36] and acid–base chemistry [37,38,39,40,41,42], resulting in second-order nonlinear partial differential equations. Under specific assumptions, these equations can be transformed into highly complex algebraic forms. For example, assuming the full dissociation of acid products and first-order enzymatic kinetics leads to oversimplified but tractable formulations [29,31]. However, merging kinetic parameters of enzymatic and protonation reactions into a single model is often unjustified, as their magnitudes differ significantly [43,44].

Algebraic simplifications are only feasible under strict assumptions such as complete substrate conversion to proteolytic products [37,38]. This scenario is equivalent to classic models for calculating the pH of weak acid/base mixtures [38]. Some studies, like those of Varanasi and colleagues [37,38], considered the role of stirring by assuming differences in species concentrations at the electrode surface and in the bulk solution.

In general, models that include the effects of stirring, inhibitors, and pH modifications become exceedingly complex. These complications arise because such models assume that transport rates are proportional to concentration gradients. This challenge can be overcome by adopting the approach proposed by Morf [45], who assumed that the transport rates were directly proportional to the species concentrations themselves. This framework produces algebraic equations analogous to kinetic equations, and naturally incorporates the impact of stirring through variations in rate constants. Morf’s model was designed for enzyme layers on potentiometric electrodes that respond to enzymatic products.

The goal of this section is to introduce a simplified but effective model for the response of pH-based potentiometric enzymatic sensors. The proposed model builds upon and modifies the substrate-enzyme electrode model developed by Morf [45].

The schematic shown in Figure 1 outlines the conceptual structure of the proposed model for a pH-based potentiometric enzymatic sensor. The substrate at concentration in the bulk solution diffuses into the enzymatic layer with a transport rate constant $k_{S}$ and may also exit at the same rate. Within the enzymatic layer, the substrate concentration decreases due to enzymatic conversion governed by Michaelis–Menten kinetics (Equation (2)). While diffusion-based models have described the behavior of enzymatic sensors responsive to the products of enzymatic reactions [29,30], these models can be adapted to pH-sensitive sensors, though at the cost of added mathematical complexity [35,39]. By assuming that transport rates are proportional to species concentrations rather than concentration gradients, the resulting equations become significantly more tractable. This simplification holds even in the presence of model extensions. In contrast, conventional diffusion models often require elaborate numerical methods unless used under highly restrictive conditions. Furthermore, the proposed model eliminates the need to define the geometry and thickness of the sensing layer. As such, it is suitable for systems in which enzymes are immobilized directly at the sensor surface such as via covalent binding or the use of dialysis membranes—scenarios where traditional diffusion models are not applicable.

Guilbault and Nagy [46] proposed an approximation method for sensors with very thin enzyme membranes, which is particularly effective when analyzing diluted substrate samples. Hameka and Rechnitz [31] attempted a mathematical treatment using Laplace transforms for potentiometric enzyme electrodes. However, they were unable to solve the most analytically relevant case, where the sensor response is dependent on the substrate concentration. Later, they introduced a more general formulation that included substrate and product concentration profiles as functions of space, though numerical solutions were still necessary [30]. Tranh-Minh and Broun [47] also adopted numerical approaches to solve the differential equations describing coupled diffusion and enzymatic reactions. In contrast, Morf [45,48] developed a simplified framework that yielded an explicit analytical expression applicable across a wide range of substrate concentrations. This theoretical foundation was further expanded by Glab and co-workers [22,23,24,25,26,27], who incorporated the effects of pH-buffering systems into the model.

Several models have been developed to describe enzymatic biosensors, focusing on diffusional [35,49], kinetic [50,51], and combined diffusional-kinetic [24,25,26,52] processes occurring on the sensor surface. The application of these models depends on the membrane thickness of the enzyme layer. For thick membranes (greater than 50 µm), diffusion becomes the limiting factor, making purely diffusional models applicable. For intermediate membrane thicknesses (10–50 µm), diffusional-kinetic models are more appropriate, as diffusion and enzyme kinetics influence the reaction. For very thin membranes (less than 10 µm), where substrate and product diffusion are rapid, kinetic models dominate as reaction rates are limited primarily by enzymatic activity. At such thin membranes, substrate and product molecules can quickly reach the enzyme and diffuse away without significant delay, minimizing the impact of diffusion. The enzymatic reaction rate (kinetics) predominantly controls the biosensor’s performance. In this case, the reaction follows Michaelis-Menten kinetics [51], where the rate is dependent on enzyme activity rather than diffusion. We checked that even after coating with PAH, our sensing layer did not exceed 1 µm (Poster S1), making the purely kinetic model perfect for our studies.

In our study, all calculations and the derivation of the main formula were performed under a simplified kinetic framework, following the approach of Glab, Koncki R., and others [24,25,26,52]. In this formulation, the exchange of species between the enzyme layer and the bulk solution is described by the transport rate constant $k_{X}$ rather than by explicit diffusion equations. This treatment assumes a steady-state condition, uniform transport constants for all protolytic species, and neglect of the detailed geometry of the enzyme layer. These assumptions are justified for thin and homogeneous PAH/urease coatings, where the concentration gradients across the film are negligible. Under such conditions, the kinetic formulation provides an accurate and computationally efficient description of the sensor response. The surface of a pH (Al/p–Si/SiO_2_/Ta_2_O_5_) sensor is coated with an enzyme that catalyzes the conversion of a substrate S into several products with their respective stoichiometric coefficients:(1)$$S+n_{x}X\overset{Enzyme}{\to }n_{A}\mathrm{HA}+n_{B}B+n_{Z}Z,$$
where *X* are other reaction substrates present, $n_{x}$ are corresponding stoichiometric coefficients; HA is protonated product *A*; *B* is the deprotonated and *Z* is the non-protolytic product. For the particular case of the urea-urease reaction, we considered here: $S=CO{\left(NH_{2}\right)}_{2}$ (Urea), $HA=H_{2}CO_{3}$, $B=NH_{3}$, $A=HCO_{3}^{-}$. The full cascade of reactions describing the enzymatic hydrolysis of urea can be represented by the following reaction equations:(2)$$CO{\left(NH_{2}\right)}_{2}+3H_{2}O\overset{Urease}{\to }2NH_{4}+HCO_{3}^{-}+OH^{-}\;H_{2}CO_{3}⇄HCO_{3}^{-}+H^{+},\;NH_{4}^{+}⇄H^{+}+NH_{3}.$$

The reaction rate V of the overall enzymatic reaction is given by the Michaelis–Menten equation [51]:(3)$$V=\frac{V_{max}\left[S\right]}{K_{M}+\left[S\right]},$$
where V is the actual reaction rate, $\left[S\right]$ is the concentration of the substrate in the enzymatic layer, $K_{M}$ is the Michaelis–Menten constant, and $V_{max}$ is the maximum reaction rate.

The scheme of Figure 1 shows the reactions and concentration diffusion between the bulk solution and enzymatic layer. The substrate (urea) concentration within bulk solution ${\left[S\right]}_{B}$ diffuses in and out of the enzymatic layer with a transport rate constant $k_{S}$. The same is true for all other species and their corresponding transport rate constant $k_{X}$, where X is the species name. The right part of the figure shows shapeless green enzyme molecules (urease) immobilized on the red line (PAH, positively charged –NH_3_^+^ groups). The background color gradient of the urease-containing layer indicates the transition from PAH to urease. A decrease in substrate concentration on the sensing layer is the result of the enzymatic reaction with the products HA and B. The products undergo continuous reversible reactions with corresponding dissociation constants $K_{aX}=\left[H,X\right]/\left[HX\right]$, where X is the corresponding pairs of substances, A is for the carbonic acid and bicarbonate, B is for ammonia and ammonium, and W is for the dihydrogen and monohydrogen phosphate ions of PBS.

Overall, for this model, we adopted the same approximations as in Ref. [24], namely:

The geometry of the sensing layer does not have to be defined in kinetic model.Enzymatic reaction takes place by Michaelis–Menten kinetics, reaction occurs under steady-state conditions (i.e., t=∞), and concentrations are constant.Dissociation constants $K_{aX}$ are the same in both layers.All species, except the enzyme, diffuse bidirectionally through the enzyme layer, with transport rate constants being the same in both directions, independent of protonation (i.e., $k_{A}=k_{HA}$, etc.) and equal to each other (i.e., $k_{H}\approx k_{W}\approx k_{S}$) and that makes normalized constants ${\bar{k}}_{W}={\bar{k}}_{H}=1$.

The rates of changes of the concentrations of substances within the enzymatic layer are represented by a set of kinetic Equations (4a)–(4e), which take into account the rates of transfer of the corresponding substances in and out of the sensitive layer as well as the enzymatic reactions:(4a)$$\frac{d\left[S\right]}{dt}=k_{S}{\left[S\right]}_{B}-k_{S}\left[S\right]-\frac{V_{max}\left[S\right]}{K_{M}+\left[S\right]}$$(4b)$$\frac{dc_{W}}{dt}=k_{W}{\left[W\right]}_{B}+k_{\mathrm{HW}}{\left[\mathrm{HW}\right]}_{B}-k_{W}\left[W\right]-k_{\mathrm{HW}}\left[\mathrm{HW}\right]$$(4c)$$\frac{dc_{A}}{dt}=n_{A}\frac{V_{\max}\left[S\right]}{K_{m}+\left[S\right]}-k_{A}\left[A\right]-k_{\mathrm{HA}}\left[\mathrm{HA}\right]$$(4d)$$\frac{dc_{B}}{dt}=n_{B}\frac{V_{\max}\left[S\right]}{K_{m}+\left[S\right]}-k_{B}\left[B\right]-k_{\mathrm{HB}}\left[\mathrm{HB}\right]$$(4e)$$\begin{array}{ll} \frac{dc_{H}}{dt}=k_{H}{\left[H\right]}_{B} & +k_{\mathrm{HW}}{\left[\mathrm{HW}\right]}_{B}+n_{A}\frac{V_{\max}\left[S\right]}{K_{m}+\left[S\right]}-k_{H}\left[H\right]-k_{\mathrm{HW}}\left[\mathrm{HW}\right]-\\ & -k_{\mathrm{HW}}\left[\mathrm{HW}\right]-k_{\mathrm{HA}}\left[\mathrm{HA}\right]-k_{\mathrm{HB}}\left[\mathrm{HB}\right] \end{array}$$
where the letters indicate the concentrations of the substances that were determined earlier. From this system of differential equations, corresponding to the scheme in Figure 1, under the steady-state approximations, an implicit functional dependence of [H] vs. [Urea] can be obtained from an algebraic sum of Equations (4b)–(4e):(5)$$\begin{array}{ll} {\bar{k}}_{H}(\left[H{]}_{B}-\left[H\right]\right) & +{\bar{k}}_{W}c_{W}^{B}\left[\frac{1}{1+\frac{K_{\mathrm{aW}}}{{\left[H\right]}_{B}}}-\frac{1}{1+\frac{K_{\mathrm{aW}}}{\left[H\right]}}\right]\\ & +\;\frac{{\bar{k}}_{V}\left[S\right]}{K_{M}+\left[S\right]}\left(\frac{n_{A}}{1+\frac{\left[H\right]}{K_{\mathrm{aA}}}}-\frac{n_{B}}{1+\frac{K_{\mathrm{aB}}}{\left[H\right]}}\right)=0 \end{array}$$(6)$$\left[S\right]=\frac{{\left[S\right]}_{B}-{\bar{k}}_{V}-K_{M}+\sqrt{{\left(K_{M}+{\bar{k}}_{V}-{\left[S\right]}_{B}\right)}^{2}+4K_{M}{\left[S\right]}_{B}}}{2}$$
where $\left[S\right]$ derived from Equation (4a), $c_{W}^{B}$ is the buffer concentration, $K_{\mathrm{aW}}$, $K_{\mathrm{aA}}$, and $K_{\mathrm{aB}}$ are the dissociation constants for the buffer and products A and B, and ${\bar{k}}_{W}$, ${\bar{k}}_{H}$*,* and ${\bar{k}}_{V}$ are the normalized constants for the transport rate and maximum reaction speed:(7)$${\bar{k}}_{H}=\frac{k_{H}}{k_{S}},\;{\bar{k}}_{W}=\frac{k_{W}}{k_{S}},\;{\bar{k}}_{V}=\frac{V_{max}}{k_{S}}.$$

The dissociation constants *K_a_* of the buffer and products are used to express protolytic species concentrations via [H^+^], and thus enter the implicit algebraic relation (Equation (5)), although they are not shown explicitly in Equations (4a)–(4e) to keep the system compact.

### 2.2. Fitting and Minimization

Thus, the model of Glab et al., through Equation (7) provides an implicit (non-explicit) dependence of the hydrogen ion H concentration on the sensing surface on the urea concentration $S_{B}$ in the bulk:(8)$$F\left(H,S_{B},K_{M},{\bar{k}}_{V},\underline{H_{B},C_{W}^{B},K_{a}^{W},K_{a}^{A},K_{a}^{B}}\right)=0$$

From here on, for the sake of simplicity, we will drop square brackets to denote concentrations. Here, H and $S_{B}$ are the variables of the problem, $K_{M},{\bar{k}}_{V}$ are the parameters that are used to minimize the residuals during the fitting procedure, and the underlined quantities are constants, known from the solution preparations or measured independently. To further simplify the numerical analysis of the model, we normalized the hydrogen ion concentration $H_{a}=\frac{H}{K_{aA}}$.


After all of these manipulations, the final expression we used in the fit to the experimental data looked like(9)$$\begin{array}{ll} \left(\frac{H_{B}}{K_{aA}}-H_{a}\right){\bar{k}}_{H}+ & \frac{c_{W}^{B}\left(\frac{H_{B}}{K_{aA}}-H_{a}\right)K_{aW}{\bar{k}}_{W}}{K_{aA}^{2}\left(H_{a}+\frac{K_{aW}}{K_{aA}}\right)\left(\frac{H_{B}}{K_{aA}}+\frac{K_{aW}}{K_{aA}}\right)}\\ & +\frac{{\bar{k}}_{V}\left(n_{A}\left(H_{a}+\frac{K_{aB}}{K_{aA}}\right)-H_{a}n_{B}\left(H_{a}+1\right)\right)S}{K_{aA}\left(H_{a}+1\right)\left(H_{a}+\frac{K_{aB}}{K_{aA}}\right)\left(K_{M}+S\right)}=0 \end{array}$$Equation (9) can be also rewritten as a fourth-order polynomial in $H_{a}$, but it will not lead us anywhere. Formally, in order to fit our model to the experiment, the solution of Equation (9)(10)$$H_{a}=f\left(S_{B},{\bar{k}}_{V},K_{M}\right),$$
should be inserted into the residual function(11)$$\Phi \left({\bar{k}}_{V},K_{M}\right)=\overset{n}{\underset{i=1}{\sum }}{\left[{\mathrm{pH}}_{a\;\exp}^{\left(i\right)}-{\mathrm{pH}}_{a}\left(S_{B}^{\left(i\right)},{\bar{k}}_{V},K_{M}\right)\right]}^{2}=\overset{n}{\underset{i=1}{\sum }}{\left[log_{10}\left[\frac{H_{a}\left(S_{B}^{\left(i\right)},{\bar{k}}_{V},K_{M}\right)}{H_{a\;\exp}^{\left(i\right)}}\right]\right]}^{2}$$
where $H_{a\;exp}$ and $H_{a}$ are the normalized experimental and theoretical concentrations of hydrogen in the sensing layer, respectively, and $pH=-\log_{10}\left[H^{+}\right]$. From the minimization procedure, the $K_{M},{\bar{k}}_{V}$ couple is found, corresponding to the best fit between the theoretical curve given by Equation (10) and the experimental points. The problem arises from the fact that in order to arrive at Equation (10), we need to insert certain values of $K_{M},{\bar{k}}_{V}$ into Equation (9) to solve it numerically. Such a step precludes the possibility of residual minimization. Glab et al. in Ref. [26] overcame this difficulty by using a simple method implemented into a domestic, unpublished PASCAL code. In order to reproduce their results, we took advantage of the Wolfram Mathematica algebra system [53] and inserted a particular couple of $K_{M},{\bar{k}}_{V}$ values into Equation (9) and solved it numerically inside the minimization procedure (see the Appendix A for the Mathematica notebook). The minimization of Equation (11) was conducted with the help of the “Sum” and “NMinimize” commands of Wolfram Mathematica. With the last command, five minimization methods are available in the basic Wolfram Mathematica package, namely: Nelder–Mead (NM), Differential Evolution (DE), Simulated Annealing (SA), Nonlinear Interior Point (NIP), and Random Search (RS). The parameter optimization was carried out within wide predefined bounds (10^−14^ − 1), and the global minimum of the objective function was taken as the final solution for each applied minimization algorithm. We found NM and DE to provide the most stable and cross-consistent results and used only these minimization methods further in the study. The NM and DE algorithms yielded almost identical parameter values; the differences obtained for the PBS and AU datasets were negligible. Therefore, a single set of parameters and corresponding errors was reported for both experiments.

## 3. Results and Discussion

### 3.1. Validation of Our Approach Against Glab et al.’s [24] Results

Using the approach as described above, we successfully reproduced the results of part 3 of the article series [28] for the urea and urease reaction. Values for constants and experimental points (digitized from Figure 1 and Figure 2) were taken from the same publication.

We chose two experiments: one showing the effect of changing the buffer concentration $c_{W}^{B}=1,5,20$ mM at constant $pH_{B}=7$, and another one showing the effect of changing the buffer’s $pH_{B}=6,7,8$ at a constant buffer concentration $c_{W}^{B}=5$ mM.

Figure 2 and Table 1 show a comparison of the fits of Glab et al. [24] with those obtained by us. A very nice correspondence between the two approaches was obvious. We attributed the slight discrepancies to the fact that Glab et al. used a code written in PASCAL, while we operated with Wolfram Mathematica. In PASCAL, the only type to describe real numbers is the “*real*”, which represents floating point numbers in single precision and has an accuracy of 6–7 significant digits, causing a large round-off error in numerical calculations. Wolfram Mathematica instead uses the “*double*” type, which has an accuracy of 16 digits and can be increased further, depending on the available memory and processing power of the machine.

### 3.2. Experimental Verification

We also checked the applicability of the proposed procedure by performing a fit to our experiments. This was the same reaction of urea and urease on a potentiometric enzymatic sensor of our own (Poster S1). Figure 3 illustrates the experimental setup of the EISCAP urea biosensor modified with PAH and urease. pH-sensitive EISCAP sensors were used as transducers. These enzyme-modified sensors detect urea via a urease-catalyzed hydrolysis reaction at or near the gate surface. The hydrolysis of urea produces OH^−^ ions (equivalently consuming H^+^ ions), thereby locally increasing the pH at the gate surface [54,55]. The generated OH^−^ ions are detected by the pH-sensitive EISCAP, resulting in a voltage-shift response that increases with the urea concentration [54]. PAH/urease-modified EISCAP urea biosensors were tested in a buffer solution of PBS (0.33 mM, pH 7.4) and artificial urine spiked with urea concentrations from 0.1 mM to 50 mM separately. This range corresponds to the expected urea levels in 10-fold diluted clinical urine samples. Urea solutions were prepared in the measurement buffer, maintaining the optimal pH for urease activity as specified by the supplier [56,57].

Various artificial urine (AU) preparation protocols exist [58,59,60]. In this study, the protocol from [3] (excluding urea) was used due to its close match with healthy human urine. For consistency with buffer-based urea measurements, AU stock solution (without urea) was diluted 10 times with deionized water and adjusted to 150 mM KCl and pH 7.4 (original pH was 6.1). Urea-spiked AU solutions (1–50 mM) were then prepared by adding urea. All reagents were from Sigma-Aldrich (Germany) and used as received. Constant capacity (CONCAP) measurements provided voltage data, which were converted to the pH values. The average value of the last hundred data points on the CONCAP plot was taken from the experimental voltage data for each urea concentration. It was assumed that approximately at that moment, the reaction reached a pseudo steady-state corresponding to the maximum reaction rate. The voltage values were converted to pH values with the assumption of linear dependence as:(12)$$pH=pH_{0}+\frac{\left(V_{0}-V_{x}\right)}{S_{pH}}$$
where $V_{x}$ is the average voltage value of the last hundred dots for urea concentration at X mM (i.e., 0.1 mM, 0.3, etc.), $pH_{0}$ and $V_{0}$ are the values for pH and voltage at the start of measurement, respectively (i.e., without urea, $S_{pH}$ is the sensor sensitivity to pH). This gives us experimental pH values that depend on the concentration of the substrate.

The artificial urine, prepared as in Ref. [17], contains ions and analytes at concentrations typical for the real human urine samples. The physiological urea levels in the human urine vary between 141 and 494 mM in undiluted urea in the case of the random urea test [56]. Urine analysis tests are often conducted by dilution of the samples, which, if diluted 10× will be exactly in the 10 mM to 50 mM range we tested in this work. The sensor exhibited a broad operational range from 0.1 mM to 50 mM urea, with sensitivities in the range of 30–35 mV dec^−1^. This wide detection range, together with sensitivity values consistent with those reported in the literature, suggests that the developed sensor is well-suited for the rapid assessment of urea concentrations under both physiological and pathological conditions.

Detailed experimental results and details of the measurements will be reported elsewhere (Appendix A); here, we will present several results of our measurements, in order to check the agreement with the theoretical model and methods.

### 3.3. Fitting Model to Our Experimental Data

Figure 4 shows four characteristic examples of measured data for the pH at the sensing layer, corresponding to some values of urea concentration in the bulk. Panels (a) and (b) correspond to the first sensor measured in PBS and AU, respectively, while panels (c) and (d) show the corresponding fittings for the second sensor under the same conditions. The extracted parameters $K_{M}$ and ${\bar{k}}_{V}$ are indicated for each fit. Since the zero concentration of substrate $S_{B}$ in the logarithm goes to infinity or undefined, it is inconvenient to show this point on the graph, so we replaced the point 0 $S_{B}$ with “0 mM>>” at (10^–5^ mM). The same code as in Section 3.1 was used for fitting, with some changes to meet the conditions of our experiments.

To evaluate the goodness of fit, the chi-square error $\left(\chi^{2}\right)$ and associated p-values were estimated for each experiment. For both the sensors and their corresponding experiments presented in Figure 4, the results confirmed the superior performance of the methods applied, yielding minimal $\chi^{2}$ values and *p*-values approaching 1, indicative of an excellent agreement between the experimental and simulated data, thus leading to reliable and chemically meaningful parameter estimates.

To support the results obtained by fitting and chi-square analysis, classical graphical linearization methods were used as an additional means of qualitative validation, specifically, the Lineweaver–Burk, Eadie–Hofstee, and Hanes–Woolf methods. As previously shown in Equation (7), the normalized parameter ${\bar{k}}_{V}$ is related to the maximal reaction rate by:$${\bar{k}}_{V}=\frac{V_{max}}{k_{S}}$$
where $k_{S}$ is the transport rate constant for the substrate. We assumed that(13)$$k_{S}=\frac{D}{l\times t}$$
with $D=\;9.3\times 10^{-5}$ cm^2^ s^−1^ as the diffusion constant of the substrate [61], l=1 cm and $t=10^{-7}$ cm are the length and thickness of the active enzyme layer. Based on this assumption, values of $V_{max}$ were obtained from the fitted ${\bar{k}}_{V}$ and $K_{M}$ and were found to be in complete agreement with the results obtained from the graphical methods (Table 2). This consistency further strengthens the reliability of our parameter estimation.

Although these graphical methods are not suitable for precise parameter estimation due to their sensitivity to experimental noise and error propagation, they remain useful for obtaining approximate kinetic parameters. The parameters were found to be so close that their curves, depicted together, were graphically indistinguishable, therefore the numbers from which the differences between the values began were underlined in Table 2. This close agreement further confirms the validity of the proposed model and the reliability of the fitting results. The close agreement between the sets of parameters further confirms the validity of the proposed model and the robustness of the fitting results.

Together, the data presented in this section validate the fitting methodology and confirm that under appropriate conditions, the proposed kinetic model is capable of accurately reproducing the experimental results for urea measurement across both simple buffer systems and physiologically relevant artificial urine matrices.

## 4. Conclusions

In this work, we applied a previously developed kinetic model [23,24,25,26,27,28] to quantitatively describe the response of a PAH/urease-functionalized EISCAP urea biosensor. The model, derived with simplifying assumptions based on Morf’s approach, provides an implicit analytical relationship between the sensor’s pH response and the urea concentration. By fitting this model to the experimental calibration data, we extracted the enzyme’s apparent Michaelis–Menten and reaction rate constants in different media. The fitted parameters for a simple buffer (PBS) and a complex artificial urine (AU) matrix were consistent with the expected urease behavior and previously reported values, demonstrating the model’s validity. The model accurately reproduced the literature results and revealed how factors like buffer strength and pH affect sensor performance.

Overall, the integration of theoretical modeling with experimental validation yielded a deeper understanding of the biosensor’s operation. The quantitative agreement between the model predictions and experimental observations across varying conditions confirms that our model captures the key mechanisms (enzyme kinetics and mass transport) governing the sensor response. Such a model can be a useful tool for sensor design, as it allows us to assess how the local pH in the enzyme layer, and therefore the concentration of the target substrate detected in the solution changes. The insights gained can guide optimization of the device (for example, tuning enzyme loading or membrane structure to achieve desired sensitivity and range). More broadly, this study illustrates that combining steady-state modeling with data fitting can greatly aid in characterizing immobilized enzymes in biosensing applications, ultimately contributing to the development of more reliable and high-performance biosensors. Last, but not least, for the first time, the applicability of a kinetic model defined with Equation (4) to describe urea detection in an artificial urine (AU) solution has been reported.

## Figures and Tables

**Figure 1 sensors-25-06596-f001:**
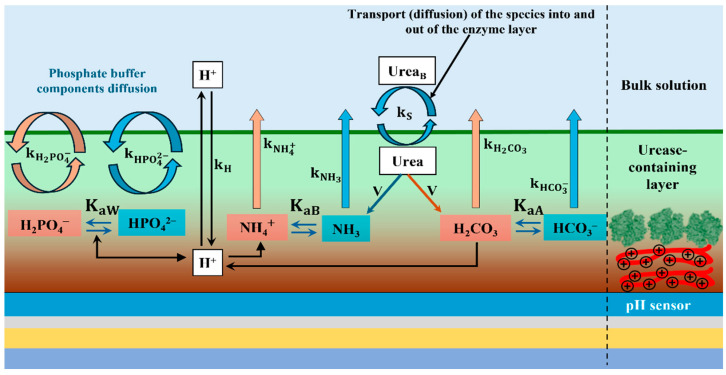
Scheme of the pH-based potentiometric enzymatic sensor. Red squares indicate the concentrations of protolytic species (acids) and blue squares are for non-protolytic species (bases). $k_{X}$ and $K_{aX}$, with corresponding subscripts, are the transport rate constants and dissociation constants, respectively, for species indicated by the subscripts. Right part illustrates the enzyme-containing layer; red line denotes the PAH film carrying positively charged –NH_3_^+^ groups; irregular green shapes represent immobilized urease molecules.

**Figure 2 sensors-25-06596-f002:**
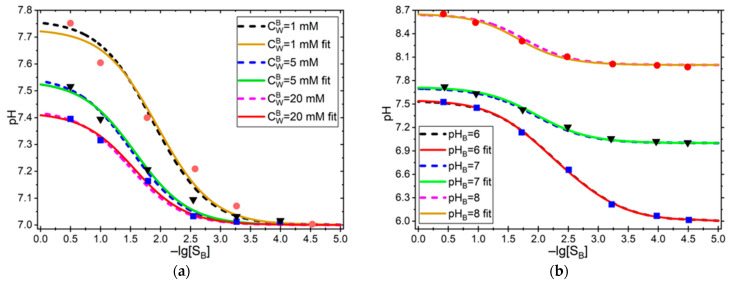
Urea response of the pH-based biosensor (**a**) at varying phosphate concentrations of the buffer shown in legend; and (**b**) for varying buffer pH in the bulk, shown in the legend. Dots are experimental points and dashed lines are fits, reproduced from Ref. [24], and solid lines show the results of our fit.

**Figure 3 sensors-25-06596-f003:**
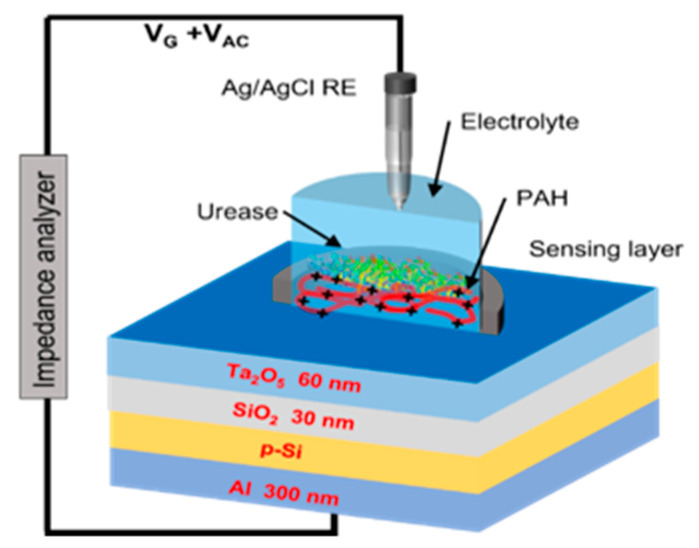
3D scheme of measurement setup.

**Figure 4 sensors-25-06596-f004:**
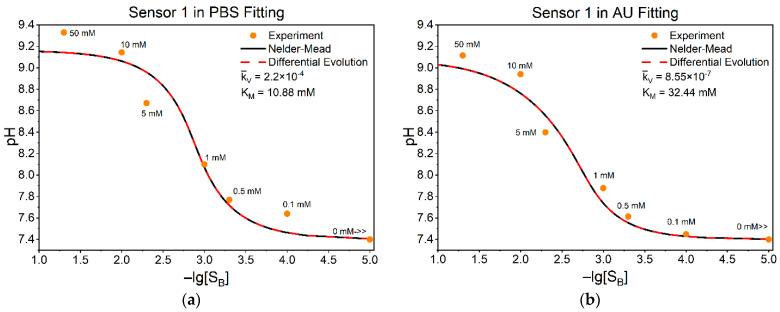
Hydrogen cation concentration at the sensing layer vs. urea concentration in the bulk in negative logarithmic scale. Panels (**a**,**b**) show fittings for sensor 1 measured in PBS and AU, respectively; panels (**c**,**d**) show fittings for sensor 2 measured in PBS and AU, respectively. “0 mM>>” is to show very small (10^−5^ mM), nearly zero concentrations of urea. Experimental points are shown in orange, and fitted Equation (10) as a solid line. Goodness of fit $\chi^{2}$: sensor 1—PBS (**a**): 12.23 × 10^−3^, AU (**b**): 11.56 × 10^−3^; sensor 2—PBS (**c**): 7.59 × 10^−3^, AU (**d**): 3.04 × 10^−3^.

**Table 1 sensors-25-06596-t001:** Comparison of our fitted parameters and those obtained in the article, with respective chi-square values of our fit. First with constant pH_B_ = 7 and changing $c_{W}^{B}$, then with constant $c_{W}^{B}=5$ mM and changing pH_B_.

Constants	Values	Results from Ref. [24]		Our Approach		
		$k_{V0}\times 10^{-3}$	$K_{M0}$ (mM)	$k_{V}\times 10^{-3}$	$K_{M}$ (mM)	$\chi^{2}\times 10^{-3}$
$c_{W}^{B}$ (mM)	1	0.41	13.1	0.39	11.18	5.33
	5	1.50	32.1	1.46	27.57	3.01
	20	4.62	33.3	4.52	28.43	0.4
$pH_{B}$	6	3.05	11.2	3.08	11.51	0.21
	7	1.90	10.7	1.93	10	2.3
	8	0.86	16.5	0.87	21.11	7.1

**Table 2 sensors-25-06596-t002:** Comparison of found parameters $V_{max}$ and $K_{M}$ found from assumption Equation (13) and using graphical methods.

Methods	PBS		AU	
	$V_{max}$, M/L × s	$K_{M}$, mM	$V_{max}$, M/L × s	$K_{M}$, mM
Our method	0.20	10.88	7.95 × 10^−4^	32.45
Lineweaver–Burk	0.20	10.88	7.95 × 10^−4^	32.45
Eadie–Hofstee	0.20	10.88	7.95 × 10^−4^	32.45
Hanes–Woolf	0.20	10.88	7.95 × 10^−4^	32.45

## Data Availability

Data will be made available upon request.

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
