# Peer review of "Urea Detection in Phosphate Buffer and Artificial Urine: A Simplified Kinetic Model of a pH-Sensitive EISCAP Urea Biosensor"

_sensors, 2025, doi:10.3390/s25216596_

Round 1

Reviewer 1 Report

Comments and Suggestions for Authors

This manuscript reports a novel biological pH sensitive detection technique for detecting urea in phosphate buffer and artificial urine. The research results of this manuscript have practical significance for the development of detection technology for urea in liquid phase. Based on the current manuscript, we suggest that the author team optimize the manuscript in the following areas.
(1) Has the data result in Figure 2 taken into account the error of multiple experiments;
(2) Will the presence of other substances that interfere with the pH value in the solution affect the detection results?
(3) Can this technology be used for urea detection in other liquid phase systems besides phosphate buffer and artificial urine?
(4) What are the advantages of the technology reported by the author compared to existing technologies?
(5) Will the ambient temperature during the detection process have an impact on the testing precision?

Author Response

General comments

First, we would like to express our deepest gratitude to all four reviewers, who carefully read our paper, and raised very relevant comments, which undoubtedly improve the content and presentation of our submission.

The considered paper was devoted to the theoretical model of the biosensor more, than to the experiment. However, from the reviewers comments we see the need to improve the description of the experimental part, which we did in the current version of the paper.

Another group of comments originated from format-related flaws, which appeared because the corresponding author used LibreOffice instead of Word. We have resolved the problem in the current version.

Below, we answer to the comments point by point.

Reviewer 1.

Comment 1. Has the data result in Figure 2 taken into account the error of multiple experiments?

Response:
We appreciate the reviewer’s insightful questions. The data used in Figure 2 was digitized from Ref. [26] in the manuscript and not a result of our measurements. Glab et al [26] do not report in their article whether points reported are a result of multiple experiments or not. The purpose of our Figure 2 is to demonstrate that our adaptation of the kinetic model reproduces their results with high accuracy. In Section 3.2, where we present our own measurements (Figures 3–4), we relied on multiple experimental runs and performed statistical evaluation (chi-square error and p-value analysis), confirming the reproducibility of our method.

Comment 2. Will the presence of other substances that interfere with the pH value in the solution affect the detection results?

Response: The presence of other substances does not significantly interfere with the pH value in the solution and, therefore, does not affect the detection results. In our experiments, the EISCAP biosensor signal is obtained as the difference between the responses of two measurement cells—one containing urea and the other serving as a reference without urea. This differential measurement approach effectively compensates for any baseline pH fluctuations or contributions from other ionic or buffering components present in the medium. As a result, only the pH variation associated with the urease-catalyzed hydrolysis of urea is reflected in the recorded signal, ensuring accurate and selective detection.

Comment 3. Can this technology be used for urea detection in other liquid phase systems besides phosphate buffer and artificial urine?

Response:
In principle, yes. However, in this work we limited ourselves to PBS and AU because (i) PBS is a standard laboratory medium for enzymatic studies, and (ii) AU closely mimics the composition of real urine and thus serves as a realistic test matrix for biomedical applications. Extension to other systems, such as blood serum or alternative synthetic buffers, is in principle possible but would require additional experimental validation and knowledge of relevant constants. In general, sensors based on the same technology, have also been used by the INB group for the measurements of the acetoin in the wine, beer, etc. [13]

Reference to measurements in different solutions added to the introduction։ ”Notably, Welden et al. employed EISCAP sensors for the measurement of acetoin in wine and beer media, demonstrating the versatility of this platform biochemical sensing applications [13].” [Lines 80-82]

Comment 4. What are the advantages of the technology reported by the author compared to existing technologies?

Response:
The EISCAP sensors are known as universal, in terms of their chemical stability in harsh chemical environments, broad pH measurement range, short response time. The pH-sensitive Ta2O5 surface provides a near-Nernstian pH sensitivity, low drift and small hysteresis, which makes this technology reliable for measurement in varying solution media. Due to high pH-sensitivity of the EISCAP sensors we used (with Ta2O5 sensitive top layer) are able to detect the minor pH changes occurring on the surface of the sensor, while the pH of the bulk solution can remain unchanged. This allows us to measure the reaction products generated in the close vicinity of the sensor and the reaction sites (in this case the enzymatic layer), which is advantage over other technologies where the detection of the pH change in the total solution is measured, thus requiring large amount of enzymes (reactants) and also long time to be able to detect the same pH change in the bulk volumes.

We amended the introduction with the detailed discussion about the advantage of the detection technology we used.[Lines 30-136]

Comment 5. Will the ambient temperature during the detection process have an impact on the testing precision?

Response:
The ambient temperature does not affect the testing precision in our experiments, as all measurements were performed under constant room temperature conditions. However the influence of temperature is a critical consideration for enzyme-based biosensors. Urease activity and the pH-sensitive transducer output are both temperature-dependent; thus, significant changes in detection temperature can impact measurement precision. In this work, measurements were carried out under controlled laboratory temperature, but for field applications it may be necessary to implement temperature compensation or control to maintain accuracy.

Reviewer 2 Report

Comments and Suggestions for Authors

"Urea detection in phosphate buffer and artificial urine: A simplified kinetic model of a pH-Sensitive EISCAP Urea Biosensor” is a research paper that presents interesting findings, but poor formatting of equations significantly decreases the quality of the manuscript and makes it difficult to read. There are areas that could benefit from improvement.

  1. The introduction section contains a lot of general information about biosensors, but there is no critical review of different types of known urease biosensors or information about gold standard methods. This information should be included.
  2. Line 185 and 194 have problems with the reaction.
  3. Line 94: the first reaction

CO(NH2)+2H2O Urease → 2NH4+ + HCO3- + OH−

No balance by substrate.

  1. Formulas 5 and 6 need to be completed.
  2. Figure 1 needs to include a substrate. What is labeled as "+"? Why is only one enzyme associated with this "+"?
  3. Formulas on lines 234 and 255 need to be checked.
  4. Why take into account the interaction between the base and the acid? In Fig. 1, we introduced KaW, KaB, and KaA, but they were not used in the kinetic scheme in formulas 4a-4e.
  5. Line 254 of the formula is incorrect.
  6. Line 395: "checked the validity" Please add quantified results from statistical estimation to support this conclusion.
  7. Line 453, the diffusion constant needs measurement units. How was the length and thickness of the active enzyme layer estimated?
  8. Lines 389-390: Please add a comment on how the model was applied to predict sensor performance based on changes in enzyme layer or environmental conditions.

This research is interesting and the manuscript has potential for publication after revision.

Author Response

General comments

First, we would like to express our deepest gratitude to all four reviewers, who carefully read our paper, and raised very relevant comments, which undoubtedly improve the content and presentation of our submission.

The considered paper was devoted to the theoretical model of the biosensor more, than to the experiment. However, from the reviewers comments we see the need to improve the description of the experimental part, which we did in the current version of the paper.

Another group of comments originated from format-related flaws, which appeared because the corresponding author used LibreOffice instead of Word. We have resolved the problem in the current version.

Below, we answer to the comments point by point.

Reviewer 2

  1. The introduction section contains a lot of general information about biosensors, but there is no critical review of different types of known urease biosensors or information about gold standard methods. This information should be included.

Response: We supplemented the introduction with a detailed discussion of the advantages of the detection technology we used and a critical review of the different types of known urease biosensors [Lines 30-136].

  1. Line 185 and 194 have problems with the reaction.

Response: Thank you, due to file transfer between our colleagues, some formulas were distorted or have missing parts. We fixed all of them (including that lines) in the new version of manuscript [Lines 240-241].

  1. Line 194: the first reaction “CO(NH2)+2H2O Urease → 2NH4+ + HCO3− + OH−“ No balance by substrate.

Response: The first reaction on line 248 was balanced by substrate, as requested. Also, we have rewritten two other reactions on line 249 in a way better corresponding to the scheme on Fig.1. [Lines 248 -249]

  1. Formulas 5 and 6 need to be completed.

Response: Thank you, due to file transfer between our colleagues, some formulas were distorted or have missing parts. We have updated the formula. [Lines 293-294]

  1. Figure 1 needs to include a substrate. What is labeled as " + "? Why is only one enzyme associated with this " + "?

Response: Fig.1 already contains the substrate (urea). The “+” refer to protonated groups of PAH (–NH₃⁺) polyelectrolyte, which serves to immobilize urease on the polymer layer. To avoid confusion, we clarified this directly in the caption and in the text (Section 2.1). We have also increased the size of text on the figure, to make it more noticeable.

Changes in text (before Fig.1): “The right part of the figure shows shapeless green enzyme molecules (urease) immobilized on the red line (PAH, positively charged –NH₃⁺ groups). The background color gradient of urease-containing layer indicates the transition from PAH to urease” [Lines 260-263].

Changes in caption: “Right part illustrates enzyme-containing layer: red line denotes PAH film carrying positively charged –NH₃⁺ groups, irregular green shapes represent immobilized urease molecules.” [Lines 273-275]

  1. Formulas on lines 234 and 255 need to be checked.

Response: Thank you, due to file transfer between our colleagues, some formulas were distorted or have missing parts. These lines contain the formulas 5 and 6, which were already mentioned previously in comment 4. We have updated the formulas. [Lines 293-294]

  1. Why take into account the interaction between the base and the acid? In Fig. 1, we introduced KaW, KaB, and KaA, but they were not used in the kinetic scheme in formulas 4a-4e.

Response: We thank the reviewer for this observation. Formulas 4a-4e contain concentrations of certain species, which enter into the expressions for the dissociation constants KaW, KaB, and KaA, as KaX = [H][X] / [HX]. We added this formula into the text right before Fig.1. They are not written explicitly in Eqs. (4a–4e) in order to keep the system compact, since those equations represent total species balances with transport terms. For clarity, we have also added an explanatory sentence after Eq. 7: “The dissociation constants Ka of buffer and products are used to express protolytic species concentrations via [H⁺], and thus enter the implicit algebraic relation (Eqs. 5) although they are not shown explicitly in Eqs. (4a–4e) to keep the system compact.”. [Lines 299-301]

  1. Line 254 of the formula is incorrect.

Response: The reviewer correctly noticed that the hydrogen ion concentration symbol [H] was missing in the small formula on line 254; we have fixed the printed formula to: pH= − log10[H+] [Line 318]

  1. Line 295: "checked the validity" Please add quantified results from statistical estimation to support this conclusion.

Response: As shown in the caption of Fig.2 “Dots are experimental points and dashed lines are fits, reproduced from Ref. 26], and solid lines show the results of our fit.” By stating, that we checked the validity, we meant that fitted curves we plotted, and the values of fitted parameters we obtained are very close to those, reported by Glab et al in Ref. [26]. Unfortunately, Glab et al did not report their χ2 values, so that we can only report ours, which have been added to Table 1. [Line 363-365]

  1. Line 453, the diffusion constant needs measurement units. How was the length and thickness of the active enzyme layer estimated?

Response: Thanks, we have added measurement units for the diffusion constant, which now reads Ds ​=9.3×10−5 cm2 s−1 (value taken from literature for small neutral solutes; Ref. [61] cited in manuscript).

Regarding the measurement of the enzyme layer thickness:

We have measured the total thickness of the PAH/Urease membrane by ellipsometry, which is in the order of 2.5 - 4.5 nm. The results of our measurements for both samples are given in the attached file (named Thickness Measurement).

We have not measured the thickness of the enzyme layer itself, and there is a need for additional detailed measurements and accurate estimates. We have not set such a problem in this article and have not made accurate measurements of the enzyme layer thickness, since the membrane thickness is not necessary for the application of the kinetic model and it works well even at nanometer thicknesses (as is described in the manuscript [Lines 221-227]), and with the approach stated in the article, we have simply estimated the values ​​of the Vmax and KM parameters by considering the enzyme layer thickness as 1 nm, and obtained results that coincide with the fitting results. If such an approach is not acceptable to the esteemed reviewer, we agree to simply remove that part from the current article.

  1. Lines 389-390: Please add a comment on how the model was applied to predict sensor performance based on changes in enzyme layer or environmental conditions.

Response: Regarding this comment, we can state that it is the result of an unsuccessful statement on our part, therefore it has been reformulated as follows: «Such a model can be a useful tool for sensor design, as it allows us to assess how the local pH in the enzyme layer and therefore the concentration of the target substrate detected in the solution changes» [475-477].

Reviewer 3 Report

Comments and Suggestions for Authors

This paper describes a simplified kinetic model for the quantitative analysis of a potentiometric, pH-based urea biosensor. The topic is relevant, and the manuscript is generally well structured. However, before the paper can be considered for publication, I have several important questions and concerns that need to be addressed:

  1. Validation with real samples
    The study relies exclusively on phosphate buffer and artificial urine. Why were no experiments conducted with real urine samples? How can the authors ensure that their kinetic model remains valid under physiologically relevant and more variable conditions?

  2. Impact of simplifying assumptions
    The proposed model is based on strong simplifications (e.g., steady-state approximation, uniform transport constants, neglect of layer geometry). How do these assumptions affect the accuracy of the model, and what are the potential limitations when applying it to more complex or thicker enzyme layers?

  3. Robustness of parameter extraction
    The fitting procedure appears highly dependent on the chosen numerical method. How sensitive are the extracted kinetic parameters (Km, Vmax, k) to the initial guesses, experimental noise, and the optimization algorithm? Please provide an error analysis or sensitivity study.

  4. Reproducibility and stability of the biosensor
    The manuscript does not provide information about sensor-to-sensor reproducibility or the operational lifetime of the PAH/urease-modified EISCAP biosensor. How reproducible are the measurements across multiple devices, and what is the expected stability over time.

  5. Although differences between buffer and artificial urine are highlighted, the clinical implications of these differences are not discussed. How does the biosensor’s sensitivity, detection limit, and dynamic range compare with existing urea detection methods, and does it meet clinically relevant concentration ranges in urine for healthy and pathological conditions?

Author Response

General comments

First, we would like to express our deepest gratitude to all four reviewers, who carefully read our paper, and raised very relevant comments, which undoubtedly improve the content and presentation of our submission.

The considered paper was devoted to the theoretical model of the biosensor more, than to the experiment. However, from the reviewers comments we see the need to improve the description of the experimental part, which we did in the current version of the paper.

Another group of comments originated from format-related flaws, which appeared because the corresponding author used LibreOffice instead of Word. We have resolved the problem in the current version.

Below, we answer to the comments point by point.

Reviewer 3

This paper describes a simplified kinetic model for the quantitative analysis of a potentiometric, pH-based urea biosensor. The topic is relevant, and the manuscript is generally well structured. However, before the paper can be considered for publication, I have several important questions and concerns that need to be addressed:

  1. Validation with real samples. The study relies exclusively on phosphate buffer and artificial urine. Why were no experiments conducted with real urine samples? How can the authors ensure that their kinetic model remains valid under physiologically relevant and more variable conditions?

Response: The sensors are firstly validated and characterized in the artificial urine buffers in order to simulate the sensor response before testing them in close-to-physiological solutions. The use of real human urine requires specific ethical approvals, biosafety clearance, and clinical collaboration agreements to comply with institutional and national regulations for handling biological materials. These procedures were not within the initial scope and timeline of the current study but are planned as a subsequent validation stage.

  1. Impact of simplifying assumptions. The proposed model is based on strong simplifications (e.g., steady-state approximation, uniform transport constants, neglect of layer geometry). How do these assumptions affect the accuracy of the model, and what are the potential limitations when applying it to more complex or thicker enzyme layers?

Response: We thank the reviewer for this comment. The present model follows the simplified kinetic approach of Glab et al., where mass transfer between the enzyme layer and the bulk solution is represented by transport rate constants kX rather than explicit diffusion equations. This formulation introduces several simplifications: (i) the steady-state assumption, (ii) the use of uniform transport constants for all protolytic species, and (iii) neglect of explicit layer geometry. The steady-state approximation is justified because each data point corresponds to a quasi-stationary signal plateau (averaged over the last 100 ConCap points). Using uniform kX values is appropriate for thin enzyme films such as our PAH/urease layer (<1 μm), where concentration gradients are negligible. For thicker or highly heterogeneous films, a diffusion–kinetic model would be required to capture spatial concentration profiles. We have changed paragraph before Eq. (1) to clarify these assumptions and define the range where this simplified kinetic description remains accurate.

Changed text: «In our study, all calculations and the derivation of the main formula were performed under a simplified kinetic framework, following the approach of Glab, Koncki R. and others [18-20,43]. In this formulation, the exchange of species between the enzyme layer and the bulk solution is described by transport rate constants kX rather than by explicit diffusion equations. This treatment assumes a steady-state condition, uniform transport constants for all protolytic species, and neglect of the detailed geometry of the enzyme layer. These assumptions are justified for thin and homogeneous PAH/urease coatings, where concentration gradients across the film are negligible. Under such conditions, the kinetic formulation provides an accurate and computationally efficient description of the sensor response». [Lines 229-238]

  1. Robustness of parameter extraction. The fitting procedure appears highly dependent on the chosen numerical method. How sensitive are the extracted kinetic parameters (KM, Vmax, kV) to the initial guesses, experimental noise, and the optimization algorithm? Please provide an error analysis or sensitivity study.

Response: We thank the reviewer for this complex question and we answer it in order.

The kinetic parameters (KM, kV) were determined according to least squares method. The parameter search was performed within broad ranges (from 10−14 to 1), and the global minimum of the sum function was selected as the final result for each minimization method mentioned in lines 329–331.

For each substrate concentration, the average of the last 100 voltage readings was taken once the sensor signal became stable, and these mean voltages were converted to pH values at corresponding concentration for experimental dots as shown in Figure 3. We believe that in this way we significantly reduce the experimental noise.

As described in lines 329–331, five independent optimization algorithm exist. We have Nelder-Mead as default method. In response to the reviewer’s comment, lines 331-338 and Figure 4 has been updated to include the Differential Evolution results as well, confirming the robustness and numerical stability of the parameter extraction procedure. The comparison of all five methods of minimization is beyond the goals of our paper. The Nelder–Mead and Differential Evolution methods produced nearly identical results: the differences between methods for the reported experiments in PBS (~10−9 for kV, ~10−7 for KM and ~10−11 for χ2) and in AU (~10−11 for kV, ~10−6 for KM and ~10−7 for χ2) are minuscule and therefore negligible. Because of that, only one set of parameters and errors was reported for both experiments.

Changes in text: ” The parameter optimization was carried out within wide predefined bounds (10−14 – 1), and the global minimum of the objective function was taken as the final solution for each applied minimization algorithm. We find NM and DE to provide the most stable and cross-consistent results and use only these minimization methods further in the paper. The NM and DE algorithms yielded almost identical parameter values; the differences obtained for PBS and AU datasets were negligible. Therefore, a single set of parameters and corresponding errors was reported for both experiments Fig.4.” [Lines 331-338]

  1. Reproducibility and stability of the biosensor

The manuscript does not provide information about sensor-to-sensor reproducibility or the operational lifetime of the PAH/urease-modified EISCAP biosensor. How reproducible are the measurements across multiple devices, and what is the expected stability over time?

Response:

We have added fittings from one more sensor under the same experimental conditions in Fig. 4. Their fitted parameters and errors are also included in caption.

Changed text: ”Fig. 4 presents four fitted pH–concentration curves obtained from the model. Panels (a) and (b) correspond to PBS and AU fittings for the same sensor, while panels (c) and (d) show fittings for two additional sensors. The corresponding KM and kV values derived from each fit are indicated.” [Lines 413-417]

“To evaluate the goodness of fit, chi-square error and the associated p-values were estimated for each experiment. For both sensors and their corresponding experiments cases presented in Fig. 4,” [Lines 428-430]

Changes in caption: “Hydrogen cation concentration at the sensing layer vs urea concentration in the bulk in negative logarithmic scale. Panels (a) and (b) show fittings for sensor 1 measured in PBS and AU, respectively; panels (c) and (d) show fittings for sensor 2 measured in PBS and AU, respectively. “0 mM>>” is to show very small (10-5 mM), nearly zero concentrations of urea. Experimental points are shown in orange, and fitted Eq. 10 as a solid line. Goodness of fit : sensor 1 — PBS (a): 12.23×10⁻³, AU (b): 11.56×10⁻³; sensor 2 — PBS (c): 7.59×10⁻³, AU (d): 3.04×10⁻³.” [Lines 422-427]

Urea Concentration (mM)

Difference with 0 mM urea (mV)

PBS

AU

Sensor 1

Sensor 2

Sensor

1

Sensor 2

0.1

13

15

3

6

0.5

20

33

12

15

1

37

42

26

25

5

70

58

55

33

10

95

67

84

43

50

105

79

93

50

As an addition, we would like to note that we agree with the respected reviewer that the article lacks consistent measurements justifying their repeatability and stability over time through measurements with a large number of different sensors. This is due to the fact that we did not put forward such a problem in the presented manuscript, but were guided by the desire to emphasize the development of a theoretical model and operational application. We plan to carry out such measurements in the upcoming research work for the simultaneous detection of three substrates (urease, creatinine, and glucose) using three enzymes (urease, creatinine deiminase, and glucose) in a multiplexed manner.

  1. Although differences between buffer and artificial urine are highlighted, the clinical implications of these differences are not discussed. How does the biosensor’s sensitivity, detection limit, and dynamic range compare with existing urea detection methods, and does it meet clinically relevant concentration ranges in urine for healthy and pathological conditions?

Response: The artificial urine, prepared as in Ref. [17], contains ions and analytes at concentration typical for the real human urine samples. The physiological urea levels in the human urine are varying between 141 - 494 mM in undiluted urea in case of the random urea test [56]. The urine analysis tests are often done by dilution of the samples, which, if diluted 10x will range in exactly the 10 mM to 50 mM range we have tested in this work. The sensor exhibited a broad operational range from 0.1 mM to 50 mM urea, with sensitivities in the range of 30–35 mV dec⁻¹. This wide detection range, together with sensitivity values consistent with those reported in the literature, suggests that the developed sensor is well suited for the rapid assessment of urea concentrations under both physiological and pathological conditions.

Changes in text: “pH-sensitive EISCAP sensors were used as transducers. These enzyme-modified sensors detect urea via a urease-catalyzed hydrolysis reaction at or near the gate surface. The hydrolysis of urea produces OH⁻ ions (equivalently consuming H⁺ ions), thereby locally increasing the pH at the gate surface [54, 55]. The generated OH⁻ ions are detected by the pH-sensitive EISCAP, resulting in a voltage-shift response that increases with the urea concentration [54]. PAH/urease–modified EISCAP urea biosensors were tested in a buffer solution of PBS (0.33 mM, pH 7.4) and artificial urine spiked with urea concentrations from 0.1 mM to 50 mM separately. This range corresponds to the expected urea levels in 10-fold diluted clinical urine samples.” [Lines 372-381]

“The artificial urine, prepared as in Ref. [17], contains ions and analytes at concentration typical for the real human urine samples. The physiological urea levels in the human urine are varying between 141 - 494 mM in undiluted urea in case of the random urea test [56]. The urine analysis tests are often done by dilution of the samples, which, if diluted 10x will range in exactly the 10 mM to 50 mM range we have tested in this work. The sensor exhibited a broad operational range from 0.1 mM to 50 mM urea, with sensitivities in the range of 30–35 mV dec⁻¹. This wide detection range, together with sensitivity values consistent with those reported in the literature, suggests that the developed sensor is well suited for the rapid assessment of urea concentrations under both physiological and pathological conditions.” [Lines 399-408]

Reviewer 4 Report

Comments and Suggestions for Authors

An excellently structured and executed research process, presented with sufficient detail and quality. After clarifying the standard deviation of the data, I would definitely recommend publication of the manuscript.

  1. What is the main question addressed by the research?

A simplified kinetic based measuring process needs to be developed and validated to enable quantitative analysis and characterization of the immobilized urease enzyme layer in a pH-based urea biosensor (EISCAP) when exposed to simple buffer and complex biological fluids such as artificial urine.Do you consider the topic original or relevant to the field?
The topic is highly relevant to the field of biosensors, particularly in clinical diagnostics and biomedical engineering. The focus on Electrolyte–Insulator–Semiconductor Capacitor (EISCAP)-based biosensors and the enzyme urease for urea detection is a common and important area of study.

2. Does it address a specific gap in the field? Please explain why this is/is not the case.

Yes, it addresses a specific gap related to the quantitative analysis and characterization of enzyme kinetics in realistic sample matrices. Standard biosensor characterization often relies on data measured in simple buffers (like PBS), which don't accurately reflect the performance, particularly the maximum reaction rate, in complex samples (like urine, blood, etc.). Inhibitory effects from components in these complex matrices can severely affect sensor performance.

3. What does it add to the subject area compared with other published material?

A validated, implicit algebraic relation derived from the steady-state approximation that links bulk urea concentration to local pH at the sensor surface. This provides a practical and reliable tool for fitting experimental data. The study demonstrates the importance of a model-based approach over mere signal measurement for accurate characterization of the immobilized enzyme layer, which is essential for translating biosensor prototypes into real-world devices.

4. What specific improvements should the authors consider regarding the methodology?

The authors should elaborate on the specific limitations of the "simplified kinetic model," especially regarding the steady-state approximation. The abstract doesn't state how well this approximation holds across the entire concentration range (0.1–50 mM) or if mass transport limitations were fully accounted for in the model derivation.

5. Are the conclusions consistent with the evidence and arguments presented and do they address the main question posed? Please also explain why this is/is not the case.

Yes, the conclusions are consistent with the evidence presented in the abstract and directly address the main question posed.

6. Are the references appropriate?

Yes, all the references are appropriate.

7. Any additional comments on the tables and figures.

I am aware that calculating the standard deviation of the parameter (or parameters) calculated in a nonlinear parameter estimation is a challenging task. Fortunately, due to the rather small amount of data to fit, I can propose an easy and quick procedure to calculate the standard deviation. The jackknife procedure is a simple and widely applicable technique for computing robust estimates of many statistics and their associated errors. It is of great value for data analysis problems in which there are no readily available techniques for calculating the standard error of a coefficient or when the analyst has reason to believe that conventional estimates of the standard error are inappropriate. In this way the quality of Table 1 and 2 can be increased.

Author Response

General comments

First, we would like to express our deepest gratitude to all four reviewers, who carefully read our paper, and raised very relevant comments, which undoubtedly improve the content and presentation of our submission.

The considered paper was devoted to the theoretical model of the biosensor more, than to the experiment. However, from the reviewers comments we see the need to improve the description of the experimental part, which we did in the current version of the paper.

Another group of comments originated from format-related flaws, which appeared because the corresponding author used LibreOffice instead of Word. We have resolved the problem in the current version.

Below, we answer to the comments point by point.

Reviewer 4

An excellently structured and executed research process, presented with sufficient detail and quality. After clarifying the standard deviation of the data, I would definitely recommend publication of the manuscript.

  1. What is the main question addressed by the research?

A simplified kinetic based measuring process needs to be developed and validated to enable quantitative analysis and characterization of the immobilized urease enzyme layer in a pH-based urea biosensor (EISCAP) when exposed to simple buffer and complex biological fluids such as artificial urine. Do you consider the topic original or relevant to the field?

The topic is highly relevant to the field of biosensors, particularly in clinical diagnostics and biomedical engineering. The focus on Electrolyte–Insulator–Semiconductor Capacitor (EISCAP)-based biosensors and the enzyme urease for urea detection is a common and important area of study.

2. Does it address a specific gap in the field? Please explain why this is/is not the case.

Yes, it addresses a specific gap related to the quantitative analysis and characterization of enzyme kinetics in realistic sample matrices. Standard biosensor characterization often relies on data measured in simple buffers (like PBS), which don't accurately reflect the performance, particularly the maximum reaction rate, in complex samples (like urine, blood, etc.). Inhibitory effects from components in these complex matrices can severely affect sensor performance.

3. What does it add to the subject area compared with other published material?

A validated, implicit algebraic relation derived from the steady-state approximation that links bulk urea concentration to local pH at the sensor surface. This provides a practical and reliable tool for fitting experimental data. The study demonstrates the importance of a model-based approach over mere signal measurement for accurate characterization of the immobilized enzyme layer, which is essential for translating biosensor prototypes into real-world devices.

4. What specific improvements should the authors consider regarding the methodology?

The authors should elaborate on the specific limitations of the "simplified kinetic model," especially regarding the steady-state approximation. The abstract doesn't state how well this approximation holds across the entire concentration range (0.1–50 mM) or if mass transport limitations were fully accounted for in the model derivation.

5. Are the conclusions consistent with the evidence and arguments presented and do they address the main question posed? Please also explain why this is/is not the case.

Yes, the conclusions are consistent with the evidence presented in the abstract and directly address the main question posed.

6. Are the references appropriate?

Yes, all the references are appropriate.

7. Any additional comments on the tables and figures.

I am aware that calculating the standard deviation of the parameter (or parameters) calculated in a nonlinear parameter estimation is a challenging task. Fortunately, due to the rather small amount of data to fit, I can propose an easy and quick procedure to calculate the standard deviation. The jackknife procedure is a simple and widely applicable technique for computing robust estimates of many statistics and their associated errors. It is of great value for data analysis problems in which there are no readily available techniques for calculating the standard error of a coefficient or when the analyst has reason to believe that conventional estimates of the standard error are inappropriate. In this way the quality of Table 1 and 2 can be increased.

Response: We feel indebted to honorable referee for his kind attitude and deep comments, which forced us to rethink the foundation of the procedures we performed. Nevertheless, we realized, that our experimental datasets were limited to 7 points, which makes jackknife analysis robust and uninformative. We have chosen to estimate the quality of fit performed using the χ2 (shown in caption of Fig.4) and p-value analysis (not shown, all values equal to 0.(9)). We hope that honorable referee will consider our estimates of quality of fit as enough to accept our results as reliable.

Round 2

Reviewer 1 Report

Comments and Suggestions for Authors

The authors have carefully revised the manuscript according to the reviewers' comments. The quality of the manuscript has been greatly improved and has reached the acceptance criteria. We recommend acceptance.

Reviewer 3 Report

Comments and Suggestions for Authors

After the central review provided by the authors, the paper is ready to go.